# Improving Object Detection for Time-Lapse Imagery Using Temporal Features in Wildlife Monitoring

**DOI:** 10.3390/s24248002

**Published:** 2024-12-14

**Authors:** Marcus Jenkins, Kirsty A. Franklin, Malcolm A. C. Nicoll, Nik C. Cole, Kevin Ruhomaun, Vikash Tatayah, Michal Mackiewicz

**Affiliations:** 1School of Computing Sciences, University of East Anglia (UEA), Norwich, NR4 7TJ, UK; m.mackiewicz@uea.ac.uk; 2School of Biological Sciences, University of East Anglia (UEA), Norwich NR4 7TJ, UK; kirsty.franklin@rspb.org.uk; 3Institute of Zoology, Zoological Society of London, Regent’s Park, London NW1 4RY, UK; malcolm.nicoll@ioz.ac.uk; 4Durrell Wildlife Conservation Trust, Les Augrès Manor, Trinity, Jersey JE3 5BP, UK; nik.cole@durrell.org; 5Mauritian Wildlife Foundation, Grannum Road, Vacoas 73418, Mauritius; vtatayah@mauritian-wildlife.org; 6National Parks and Conservation Service (Government of Mauritius), Ministry of Agro-Industry, Food Security, Blue Economy and Fisheries, Head Office, Reduit, Mauritius; kruhomaunster@gmail.com

**Keywords:** YOLO, object detection, time-lapse imagery, camera-trap imagery, temporal features, spatio-temporal features, wildlife monitoring

## Abstract

Monitoring animal populations is crucial for assessing the health of ecosystems. Traditional methods, which require extensive fieldwork, are increasingly being supplemented by time-lapse camera-trap imagery combined with an automatic analysis of the image data. The latter usually involves some object detector aimed at detecting relevant targets (commonly animals) in each image, followed by some postprocessing to gather activity and population data. In this paper, we show that the performance of an object detector in a single frame of a time-lapse sequence can be improved by including spatio-temporal features from the prior frames. We propose a method that leverages temporal information by integrating two additional spatial feature channels which capture stationary and non-stationary elements of the scene and consequently improve scene understanding and reduce the number of stationary false positives. The proposed technique achieves a significant improvement of 24% in mean average precision (mAP@0.05:0.95) over the baseline (temporal feature-free, single frame) object detector on a large dataset of breeding tropical seabirds. We envisage our method will be widely applicable to other wildlife monitoring applications that use time-lapse imaging.

## 1. Introduction

By capturing images at regular intervals, a time-lapse camera gathers data of a scene over time without the need for large quantities of video data. This makes time-lapse imaging particularly useful for applications such as wildlife monitoring, where the aim is to monitor sites over long periods of time. This presents a unique challenge for object detection, however, since the loss of temporal continuity, coupled with significant changes in illumination, makes object tracking unsuitable. In this paper, we explore methods exploiting the sequential and static nature of time-lapse imagery. These methods utilise temporal features and thereby improve scene understanding and reduce the number of false positives in the static background. Our most significant contribution is our method of temporal feature engineering for time-lapse imagery. In this method, we inject two additional spatial feature channels that capture information of stationary scenery and of non-stationary scenery. Furthermore, we demonstrate that additional improvements can be achieved using two different methods of input channel weighting. As a final contribution, we introduce a method of stratified subset sampling for object detection datasets from camera-trap imagery.

For our tests, we used a camera-trap dataset of breeding tropical ground-nesting seabirds. This consisted of approximately 180,000 images taken at various nesting locations on Round Island (RI), around 4500 of which were labelled using bounding-box annotations with the classes “Adult”, “Chick”, and “Egg”. The images (see Figure 1) were captured across 10 camera traps, each monitoring a separate scene consisting of several nesting sites. We provide more details on the dataset in Section 3. For brevity, we refer to this dataset as the RI petrel dataset.

The rest of the paper is organised as follows: first, we explore related work in wildlife monitoring and object detection in time-lapse imagery. Next, we present our proposed methods in Section 2. This is followed by a detailed description of our dataset in Section 3. We then outline our experimental procedures and findings in Section 4. Finally, in Section 5, we finish with a discussion and analysis of our results.

### 1.1. Related Work

#### 1.1.1. Deep-Learning in Wildlife Monitoring

Applying deep-learning methods to camera-trap imagery in the context of wildlife monitoring has been explored in several studies [1,2,3,4,5]. Norouzzadeh et al. [1] evaluated various Convolutional Neural Networks (CNNs) for the detection of Tanzanian wildlife in camera-trap imagery and reported that VGG [6], a deep CNN proposed by the Visual Geometry Group, yielded the highest performance. Instead of employing bounding-box predictions, their approach directly predicted the animal species and its count, limiting detections to one class of animal per image. Furthermore, they incorporated an initial stage to predict whether an animal was present before proceeding to the classification and counting phases. Upon revisiting this work in 2020, ref. [3] proposed Faster-RCNN as a more effective solution. They argued that the approaches described by [1] were overly dependent on background features due to the absence of bounding-box predictions, which aided in focusing feature learning on objects rather than the surrounding background scenery.

Hentati-Sundberg et al. [4] used YOLOv5 to gather data of seabird populations from live CCTV streams. The authors collected population counts of adult seabirds and estimated the rates of growth of chicks using the mean of predicted box sizes over time. Additionally, they detected predatory disturbances, defined as a drop in count of four or more within a one-minute period. Vecvanags et al. [5] utilised Faster R-CNN and RetinaNet to monitor populations of deer and wild boar.

#### 1.1.2. Object Detection in Time-Lapse Imagery

There is limited research on the incorporation of temporal information as features for object detection models applied to time-lapse imagery. In the context of video data, object tracking algorithms such as SORT [7] are typically employed; however, the lack of temporal continuity of time-lapse imagery renders object tracking unsuitable. A notable study relevant to our research is that of Bjerge et al. [8], where the object detection of moving targets is enhanced for image sequences. In their approach, the previous image in a sequence is subtracted from the current image, and the resulting absolute difference in each colour channel is used as a motion likelihood. This motion likelihood is then incorporated into the current RGB input through element-wise addition, where pixel values across the RGB channels are summed to produce a motion-enhanced image.

## 2. Method

In this section, we describe our technical contributions and the methods that we use. We start with a short introduction to the You Only Look Once (YOLO) object detection architecture in Section 2.1. The following Section 2.2 describes the primary contribution of our work where we detail our methods of fusing temporal information present in the time-lapse imagery sequence with the usual input of the object detector as extra input channel(s). Finally, in Section 2.3, we describe a new stratified sampling method for partitioning data into training/validation/test sets which is particularly suitable for object detection datasets with high class and annotation imbalances such as the one we used in this work.

To encourage future research or application of our methods, we have made the code available for download on GitHub (https://github.com/MarcusJenkins01/yolov7-temporal, accessed on 9 December 2024).

### 2.1. YOLOv7

YOLOv7 [9] is a single-stage object detector, where region proposal and classification are combined into joint bounding-box and class predictions. To do so, YOLOv7 consists of a number of anchor boxes for each region of the image at a number of scales. These anchor boxes are predetermined using k-means clustering to establish the mean size and aspect ratio of objects for each region in each scale. Instead of making a direct prediction for the position and size of the bounding box, the position and size is predicted as a relative adjustment of the best-fitting anchor box. By using anchor boxes and multiple scales, the predictions are kept as small adjustments to the anchor box, despite variations in object sizes and shapes; this means gradients are kept low, providing greater stability and ease of learning [10]. Of the YOLO family, we chose YOLOv7, since it was well established. Further details on the configuration of the YOLOv7 architecture we used is given in Section 4.

### 2.2. Temporal Feature Engineering

Object detectors such as YOLO usually operate with a single RGB frame as input. Here, we aim to inject temporal information into the input of the object detector as additional input channels. To develop these temporal features, we derived inspiration from background (BG) subtraction techniques. We first computed a BG model, which was then used with the current image to calculate the difference mask (DM). Both the BG model and the DM formed separate channels, which were stacked on top of the three RGB channels. Unlike [8], where the difference mask was added element-wise to the RGB input, we did not modify the RGB input, and so these features were preserved. The following subsections describe our proposed approach in greater detail.

#### 2.2.1. Temporal Average Background Model

To obtain a background model for a current image, we selected 12 prior images, from which a pixelwise mean average was computed for each of the RGB channels that was then converted to greyscale. Since images during the day and images during the night were separate modalities, the background model was separated for day and night. In other words, if the current image was taken during the day, 12 prior daytime images were selected, and likewise for nighttime imagery. This was referred to as the temporal average background model.

The set of images for daytime *D*, or nighttime *N*, for all images up to *n*, was defined as: (1)$$S^{L}=\left\{I_{i}^{L}|i<n,L\in \left\{D,N\right\}\right\}$$

$S_{B}^{L}$ is the subset of $S^{L}$ that was used to calculate the temporal average: (2)$$S_{B}^{L}=\left\{I_{j_{1}}^{L},I_{j_{2}}^{L},\ldots ,I_{j_{12}}^{L}|j_{1},j_{2},\ldots ,j_{12}\;\mathrm{are}\;\mathrm{the}\;12\;\mathrm{largest}\;\mathrm{indices}\;\mathrm{in}\;S^{L}\right\}$$

The RGB temporal average, $T_{A_{12}RGB}$, for the set $S_{B}^{L}$ was therefore given as: (3)$$T_{A_{12}RGB}=\frac{1}{12}\sum_{k=1}^{12}I_{j_{k}}^{L}$$

And the flattened temporal average, $T_{A_{12}}$, was obtained using luminosity greyscale conversion as:(4)$$T_{A_{12}}=0.299\cdot T_{A_{12}R}+0.587\cdot T_{A_{12}G}+0.114\cdot T_{A_{12}B}$$

#### 2.2.2. Difference Mask

Since we would like the network to focus on differences between the *I* and $T_{A_{12}RGB}$ that pertain to motion rather than changes in colour distribution (due to weather or lighting geometry changes), we first performed colour correction on $T_{A_{12}RGB}$ before computing the difference mask $D_{M}$.

$T_{A_{12}RGB}$ and *I* were reshaped to dimensions N×3, where N=H×W, and *H* and *W* are the image height and width, respectively. A 3 by 3 colour correction matrix, M, was then computed using least squares regression [11] as: (5)$$M=\arg\min_{M}{∥I-T_{A_{12}RGB}M∥}^{2}$$

Each pixel in $T_{A_{12}RGB}$ was then colour corrected using *M*, and the result of this operation was denoted as $T_{A_{12}RGB}^{\prime }$.

Finally, the difference mask $D_{M}$ was calculated as the absolute difference between *I* and $T_{A_{12}RGB}^{\prime }$ followed by flattening to greyscale as: (6)$$D_{M}=\sum_{k\in \{R,G,B\}}\frac{|I_{k}-T_{A_{12}k}^{\prime }|}{3}$$

The effect of applying this colour correction on $D_{M}$ can be observed in Figure 2. We can see that $D_{M}$ obtained from colour-corrected $T_{A_{12}RGB}^{\prime }$ highlights less of the stationary background scenery compared to $T_{A_{12}RGB}$ (denoted by the reduction in greyscale intensity in the background regions of the image).

#### 2.2.3. Temporal Channel Weighting

Rather than simply passing $T_{A_{12}}$ and $D_{M}$ as two additional channels alongside the RGB channels, we also trialled two techniques that applied a learned weighting to the channels $T_{A_{12}}$ and $D_{M}$. Our hypothesis was that scaling these feature channels with learned parameters before passing them to YOLOv7 would facilitate convergence toward a better local optimum. While the weightings of these channels could be implicitly learned as part of the CNN layers, we believed that explicitly scaling these channels would provide a clearer gradient flow to amplify or suppress the contribution of each of the two new feature channels. This hypothesis was confirmed in our results in Section 5. For the first method, we proposed a fixed weighting that was learned for each channel, regardless of the input values. The weightings were defined as:(7)$$W_{T_{A}}=\sigma \left(\alpha \right),\;W_{D_{M}}=\sigma \left(\beta \right)$$
where σ is the Sigmoid function, and α and β are learnable parameters. Back-propagation and optimisation of these parameters was performed end-to-end using YOLOv7’s optimiser.

For the second method, an input-aware approach of calculating weightings was also trialled using a modification of the Squeeze-and-Excitation block [12] (Figure 3). Unlike the traditional Squeeze-and-Excitation block, which applies a scale to all channels, we modified $F_{ex}$ (Equation (Equation 8)) to produce 2 weightings, which were then applied to the channels for $T_{A_{12}}$ and $D_{M}$.
(8)$$F_{ex}\left(z\right)=\sigma \left(W_{2}\delta \left(W_{1}z\right)\right),\;\mathrm{where}\;W_{1}\in R^{C\times C}\;\mathrm{and}\;W_{2}\in R^{2\times C}$$
where δ denotes the ReLU function and *C* is the number of input channels, thus C=5.

### 2.3. Subset Sampling for Training, Validation, and Test Splits

The splitting of object detection datasets into training, validation, and test subsets is often performed using random sampling. Instead, we propose a new method of stratified sampling for camera-trap imagery that ensured that each subset contained cameras with examples of minority classes, where random sampling may potentially miss these [13]. The benefit was also a trained model that was theoretically optimised for a more realistic class distribution, and similarly, a test set that was more representative.

Defining strata for object detection is complex due to the presence of multiple objects per image of varying sizes and at different locations. When we originally devised our method, there was no current research to our knowledge, but a method has since been published [14]. Analogously to our method, they used the frequency of each class in the image and the distribution of box sizes but with an explicit focus on box aspect ratios (due to the bias of aspect ratios imposed by anchor boxes for anchor-based object detection).

Since the aim of using automated methods for wildlife monitoring is often for it to be applicable to new, future cameras (scenes) at other nesting locations, we split our dataset by camera. The task was therefore to obtain a model that generalised well to unseen scenes. The cameras for each set were chosen using a combinatorial approach, where the summed variance of the class distribution, the number of objects of each predefined size, and the ratio of each class across day and night were minimised between each subset. The class distribution was computed as the mean number of objects of each class per image for each camera. The object sizes were assessed among three distinct groups, which were obtained using k-medoid (PAM) clustering with a *k* value of 3. A bounding-box label was matched with the size based on the closest cluster centre; these three sizes were interpreted as “small”, “medium”, and “large” object sizes. We used K-medoids over k-means to reduce the influence of outliers in object size on the cluster centres.

Therefore, we were looking for a partition of a set of all cameras *C*, into three subsets $C_{1}$, $C_{2}$, and $C_{3}$, where ${⋃}_{i=1}^{3}C_{i}=C$ and $C_{i}\cap C_{j}=\emptyset ,i\neq j$. Hence, we performed the following optimisation: (9)$$\min_{C_{1},C_{2},C_{3}}\left(\sigma_{N}^{2}+\sigma_{S}^{2}+\sigma_{R}^{2}\right),$$
where $\sigma_{N}^{2}$ is the sum of variances of the number of objects of each class per image among the three subsets, and *M* is the set of classes (object categories): (10)$$\sigma_{N}^{2}=\sum_{i=1}^{\left|M\right|}\sigma_{N_{M_{i}}}^{2}$$

$\sigma_{S}^{2}$ is the sum of variances of the number of objects of each object category *M* of each size category *P*, per image, among the three subsets: (11)$$\sigma_{S}^{2}=\sum_{i=1}^{\left|M\right|}\sum_{j=1}^{\left|P\right|}\sigma_{S_{M_{i}P_{j}}}^{2}$$

$\sigma_{R}^{2}$ is the sum of variances of the ratio of the number of objects of each class (object category) between day and night among the three subsets: (12)$$\sigma_{R}^{2}=\sum_{i=1}^{\left|M\right|}\sigma_{R_{M_{i}}}^{2}$$

## 3. Dataset

The RI petrel dataset was made available as part of the long-term Round Island petrel research program. This dataset was collected to monitor the breeding population of Pterodroma petrels (known locally as the “Round Island petrel”) on the Round Island Nature Reserve, a small island off the north coast of Mauritius. To obtain these data, 10 Reconyx camera traps (manufactured in Holmen, WI, USA) were deployed at 10 different nesting locations (5 Hyperfire HC600 cameras and 5 Hyperfire 2 HF2X cameras). Each camera captured the contents of between two and five petrel nests and were configured to take an image at hourly intervals, day and night, between 4 December 2019 and 8 March 2022. As outlined in our introduction, the dataset consisted of 181,635 images; of these, 4483 were labelled at the University of East Anglia using bounding-box annotations. These annotations were aided by earlier citizen-science point annotations generated through the Zooniverse citizen-science project, Seabird Watch. For more information on camera deployments and citizen-science annotations, see [15].

The nesting sites captured by the 10 cameras can be seen in Figure 4. The provided example images were taken during the day; however, during hours of low light and/or darkness (between approximately 6 P.M. and 6 A.M.), images were captured using the complementary infrared sensor.

The dataset images were annotated with the classes “Adult”, “Chick”, and “Egg” for the identification of trends in population numbers and breeding activity. Table 1 and Table 2 and Figure 5 illustrate the annotation statistics for each camera in the dataset. From these figures, it is evident that there was significant variation in class distribution, object sizes, and examples across day and night modalities between different cameras. This variability demonstrated the necessity for our method of stratification (Section 2.3) to ensure a balanced selection of cameras across the training, validation, and test sets.

Likewise, there was considerable variation in the number of images per camera. Originally, the Seabird Watch project provided 10,917 images with point-based annotations. However, when converting these to bounding-box annotations, some cameras posed greater challenges in confirming the presence and bounding area of the birds. Consequently, to ensure annotation accuracy and provide sufficient samples for a robust model, certain cameras had more annotated images due to the clearer visibility of birds compared to others.

Figure 5 shows that our annotated dataset was skewed towards a relatively constrained range of sizes for each class. Beyond that range, there were a number of outlying, larger instances for the classes “Adult” and “Chick”. These outliers typically represented a scenario where the adult or chick was in close proximity to the trap camera.

## 4. Experiments

In this section, we describe the experiments we conducted to validate the utility of the proposed methods. We start with the description of the YOLOv7 model and training configuration in Section 4.1 and Section 4.2. We describe how we use our method of stratified sampling to partition our data into training, validation, and test splits in Section 4.3. In Section A.1, we describe how we tuned the hyperparameters of the proposed models. We give details on how we performed data augmentation in Section 4.4. Finally, our main experiments evaluating the proposed methods are described in Section 4.5.

### 4.1. YOLOv7 Model Configuration

YOLOv7 offers a number of configurations with varying complexities, the most complex being YOLOv7-E6E. As model complexity increases, performance increases for the MS COCO dataset, albeit at the cost of computation time [9]. For the RI petrel dataset, images were to be analysed post-capture, not in real time. On the other hand, we still chose a balance between computation and performance due to GPU training and inference times. We opted for a middle ground between YOLOv7 and YOLOv7-E6E, YOLOv7-W6, which obtains an average precision of 54.9% on MS COCO with an inference time of 7.6 ms with an NVIDIA V100. YOLOv7-W6 is also the smallest configuration which produces bounding-box predictions for four scales, rather than the three scales of the lesser models; this makes it more robust to variation in object sizes.

### 4.2. Training Configuration

Training was performed using an RTX 6000 with 24 GB of available VRAM; thus, we used a batch size of eight for all of our experiments. For every training epoch, YOLOv7 evaluated the performance on the validation set using the “fitness score”. This was computed as: (13)fitness=0.1·mAP@0.5+0.9·mAP@0.05:0.95

Rather than using early stopping, the set of weights that obtained the greatest fitness score was stored; training was performed for the full number of epochs regardless of evaluation performance. Evaluation on the test set was then performed using these optimal weights.

### 4.3. Training, Validation, and Test Splits

Using our method of subset sampling, described in Section 2.3, we obtained the partition of our dataset shown in Table 3.

We constrained the required dataset partition to have two cameras in the validation and test sets each and six cameras in the training set. Given camera SWC4 had only 14 annotated images, we forced it to be in the training set to minimise other (single) camera bias in the validation and test sets.

We also enforced a maximum number of images for the test and validation sets, where if either of these sets exceeded an image count of 25% of the full dataset image count, the respective dataset partition was rejected.

The variables minimised during the optimisation for subset sampling are displayed in Figure 6, Figure 7, Figure 8 and Figure 9. The counts of each class and the size of each class are normalised by the number of images in each set. Our optimisation function for dataset partitioning can be thought of as minimising the sum of variances of the y-values for each x-value of these plots.

We can see that the distribution of classes, object sizes of each class, and the day and night ratios were relatively consistent across the training, validation, and test sets.

### 4.4. Data Augmentation and Baseline Model

The official implementation of YOLOv7, made available by Wang et al. [9], provides the optimal data augmentation configuration for MS COCO. For object detection, this consists of mosaic, random perspective, MixUp, HSV augmentation, and horizontal flipping (Figure A1). We followed the methodology used by Bochkovskiy et al. [16] for establishing the optimum data augmentation settings. We tested each of the methods proposed for COCO in Wang et al. [9] separately, then tested each in conjunction with each other one at a time, starting with the best performing method.

We did not optimise the individual hyperparameters for each data augmentation method, due to the large potential search space. Instead, we used the values optimised for MS COCO, with the purpose of evaluating the effect of different combinations of these methods. To that end, we used the same technique as used in hyperparameter optimisation with a 50% training subset and 50 epochs for training.

Similarly to our hyperparameters, the optimal data augmentation configuration developed for MS COCO proved to be the most effective for our dataset as well (this is the configuration illustrated in Figure A1).

To establish our baseline model (single RGB image object detector), we trained the optimal data augmentation configuration and optimal hyperparameter configuration (Section A.1) for the full number of epochs, 300.

### 4.5. Temporal Feature Engineering

To accommodate the additional channels of $T_{A_{12}}$ and $D_{M}$ and the channel weightings, a number of changes were made to YOLOv7. This predominantly included adaptations to HSV augmentation and the input layer. For training, we used the same hyperparameters as used for our baseline model and trained for 300 epochs.

#### 4.5.1. Data Augmentation of $T_{A_{12}}$ and $D_{M}$

For our experiments using channels $T_{A_{12}}$ and $D_{M}$, we applied the same data augmentation methods as those used for the RGB inputs in the baseline model, with the exception of HSV augmentation. HSV augmentation was applied to the RGB channels in the same way; however, for the $T_{A_{12}}$ channel, which was a greyscale one, only the value gain was applied. This gain was the same gain that was used for the RGB channels. We did not apply any HSV augmentation to $D_{M}$ to ensure that the difference intensity was fully preserved.

#### 4.5.2. Channel Weighting for $T_{A_{12}}$ and $D_{M}$

For both methods of channel weighting, *W* and $S_{E}$, we introduced an additional layer to the backbone, which was positioned as the first layer (before the *ReOrg* layer). The input, *x*, consisted of all channels, except the weighting was only applied to channels $T_{A_{12}}$ and $D_{M}$, and the RGB channels remained unaltered. *ReOrg* was modified to accept five channels rather than three; all subsequent layers, however, were unchanged.

#### 4.5.3. Ablation Experiments

Using the same training configuration, we trained additional models where we only provided $T_{A_{12}}$, and only $T_{A_{12}}$ and $D_{M}$ (without channel weighting). The results of these demonstrated the significance of these feature channels and the impact of channel weighting. We discuss the results of our ablation experiments in Section 5. When only $T_{A_{12}}$ was provided, we used the same HSV augmentation method as when only the value gain was applied to $T_{A_{12}}$.

## 5. Results and Discussion

The results of our experiments are shown in Table 4 and Table 5. Figure 10 and Figure 11 show a gallery of detection results for sample images from the test set.

By providing the channels $T_{A_{12}}$ and $D_{M}$, and applying a learnable weighting, we observed a significant improvement in the performance of YOLOv7 over the baseline method. We theorise that the $T_{A_{12}}$ allows YOLOv7 to exploit features of the stationary background scenery, and the $D_{M}$ channel allows regions of change to be understood. Therefore, our best model can learn to suppress detection confidence for stationary background scenery, while simultaneously leveraging the motion information offered by channel $D_{M}$ for detecting birds. This is illustrated well in Figure 10 and Figure 11, where regions of background were misclassified as birds by the baseline model but were not identified by $T_{A_{12}}+D_{M}+W$ within the threshold of 0.25. Building on this hypothesis, we can attribute the major increase in the detection performance of “Chicks” to their class resemblance to rocks in the background (due to their grey colour and rounded shape) that was closer than any other class.

$T_{A_{12}}$ + $D_{M}$ + $S_{E}$ was the most complex method, and we can see that it achieved the best mAP value on the validation set but performed worse than $T_{A_{12}}$ + $D_{M}$ + *W* on the test set. We believe this happened due to overfitting, which was made more likely by the increased number of learnable parameters in the model (complexity). In future experiments, it would be beneficial to try stronger regularisation for this method.

### 5.1. Comparison of Computational Cost

As detailed in Section 4.2, we used an RTX 6000 with a batch size of eight for all experiments. The chosen image resolution was 1280 × 1280. Table 6 illustrates the training and inference times, and the GPU memory consumed during training.

Our best method, $T_{A_{12}}$ + $D_{M}$ + *W*, resulted in a 21.8% increase in training time and a 38.1% increase in inference time. However, we believe that that increase in computational cost was justified by the 24% improvement in mAP@0.05:0.95.

### 5.2. Learned Channel Weighting

Both the fixed weightings and Squeeze-and-Excitation provided an improvement. For the fixed weightings, the learned weighting for $T_{A_{12}}$, σ(α), was 0.288, and σ(β) for $D_{M}$ was 0.824.

The performance improvement when applying such weightings could imply that the features offered by channel $D_{M}$ were immediately more distinguishing for birds than those of $T_{A_{12}}$—this was also confirmed by the visual inspection of the two channels (see Figure 12b,c)—and so this weighting allowed for a better local optimum to be converged towards earlier in the training. The fact neither weighting cancelled either channel out also further demonstrated that both these channels were useful, in addition to the evidence provided in Table 4.

### 5.3. Explicit Versus Implicit Difference

We theorised that the difference mask $D_{M}$ may not be needed, since the difference between $T_{A_{12}}$ and the RGB image can be learned implicitly. We can see, however, that providing $D_{M}$ offered an improvement of 3.8%. Perhaps by providing $D_{M}$, the effort of learning this change was minimised, and so more model resources were available for improving learning of other features. In addition, since more salient features were immediately present, the path of optimisation towards the more relevant local minima of the loss function was perhaps more stable and easier to follow.

## 6. Conclusions

In this paper, we introduced two innovative methodologies aimed at improving object detection in time-lapse camera-trap imagery, which is critical for ecological monitoring of animal populations. In our primary contribution, we leveraged temporal information to significantly enhance object detection model accuracy. By integrating features that distinguish static and dynamic elements within the input image, we achieved a notable improvement of 24% in mean average precision over the baseline.

Our secondary contribution, a method of stratified dataset subset selection, presented a novel approach to partition time-lapse imagery object detection datasets. The method ensured a balanced representation of various cameras across the training, validation, and test sets, with the aim of providing a model that generalised well across various classes, different object sizes, and day and night modalities and where the validation/test set evaluation metrics were indicative of the future model performance on unseen data.

## Figures and Tables

**Figure 1 sensors-24-08002-f001:**
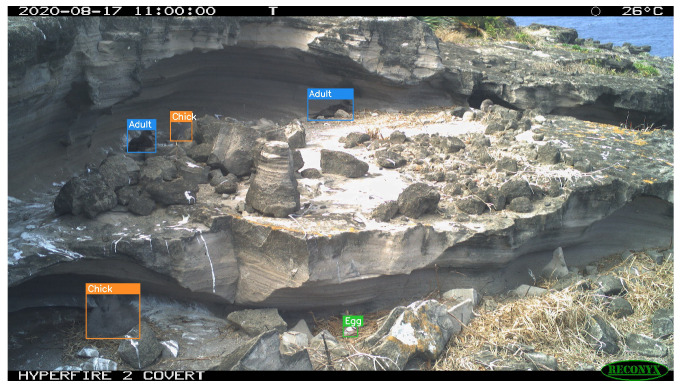
An example annotated image from the RI petrel dataset.

**Figure 2 sensors-24-08002-f002:**
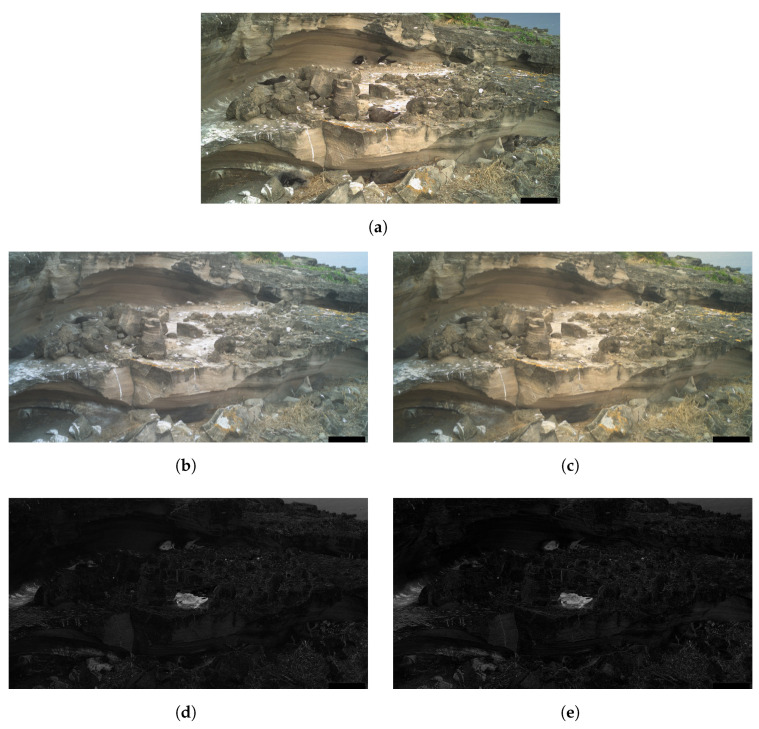
Comparison of the effect of colour correction on the difference mask, $D_{M}$. (**a**) Sample image from camera SWC3. (**b**) Corresponding $T_{A_{12}RGB}$ (before colour correction). (**c**) Corresponding $T_{A_{12}RGB}^{\prime }$ (after colour correction). (**d**) $D_{M}$ using uncorrected $T_{A_{12}RGB}$. (**e**) $D_{M}$ using colour-corrected $T_{A_{12}RGB}^{\prime }$.

**Figure 3 sensors-24-08002-f003:**
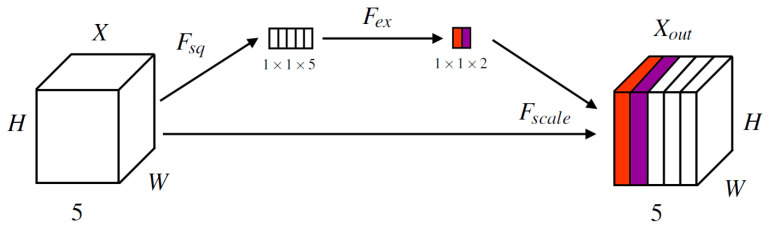
Modified Squeeze-and-Excitation block for input-aware $T_{A_{12}}$ and $D_{M}$ channel weightings. Input *X* is the output of two convolutional layers with a kernel size of 3 × 3 and stride 1 × 1, with an intermediate ReLU layer. For $F_{sq}$, global average pooling is used across the channel dimension of *X*, and $F_{ex}$ is a feed-forward network with a sigmoid output layer (to produce a scaling for each channel between 0 and 1). $F_{scale}$ denotes the multiplication between the output of $F_{ex}$ and the input channels *X* to give $X_{out}$.

**Figure 4 sensors-24-08002-f004:**
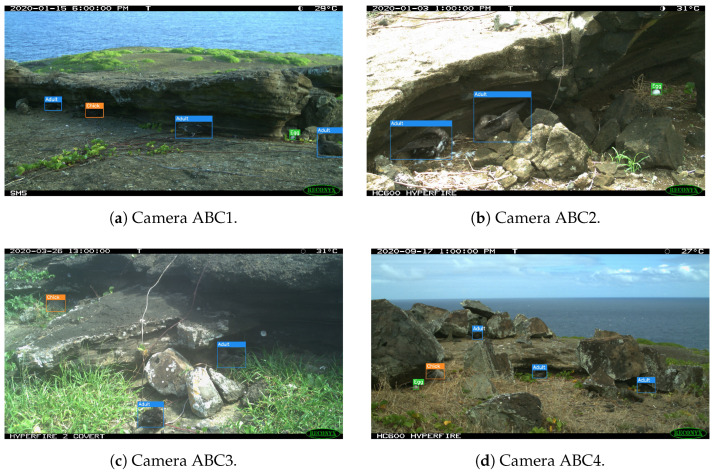
Sample images from the 10 cameras that comprised our dataset.

**Figure 5 sensors-24-08002-f005:**
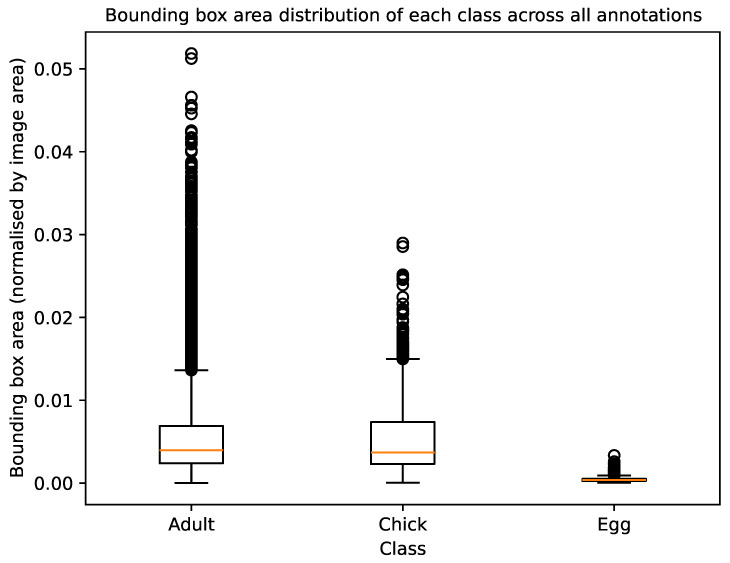
Box plots depicting bounding-box area distribution for each object category, where the area is normalised by the respective image’s dimensions.

**Figure 6 sensors-24-08002-f006:**
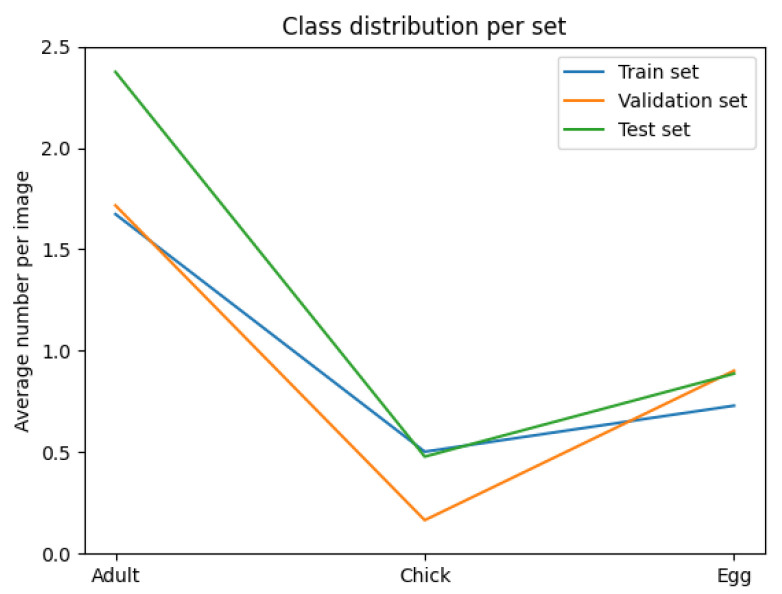
Class occurrence across each set (normalised by image count).

**Figure 7 sensors-24-08002-f007:**
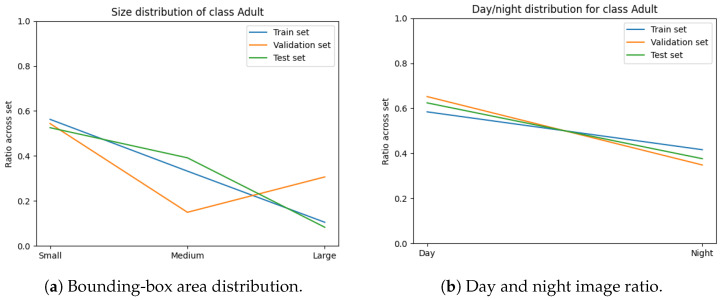
Distribution of class “Adult” across each set.

**Figure 8 sensors-24-08002-f008:**
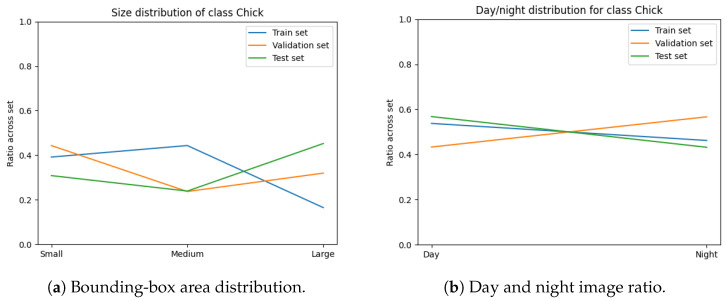
Distribution of class “Chick” across each set.

**Figure 9 sensors-24-08002-f009:**
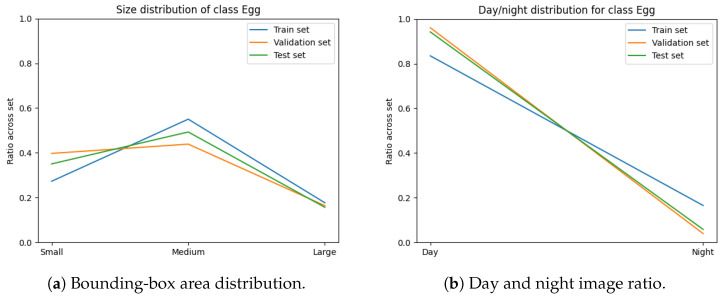
Distribution of class “Egg” across each set.

**Figure 10 sensors-24-08002-f010:**
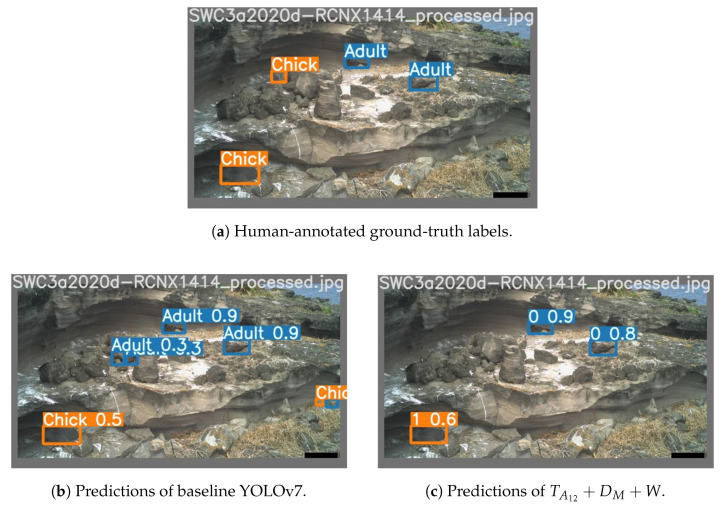
Visualisation of predictions during the day with a confidence threshold of 0.25.

**Figure 11 sensors-24-08002-f011:**
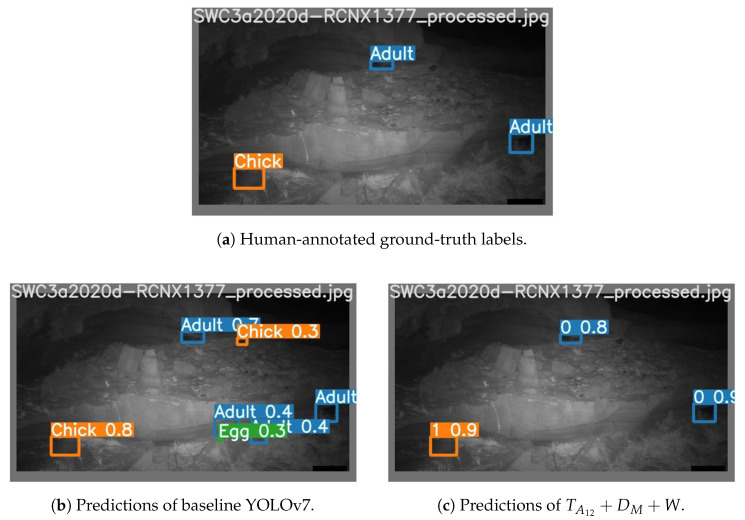
Visualisation of predictions during the night with a confidence threshold of 0.25.

**Figure 12 sensors-24-08002-f012:**
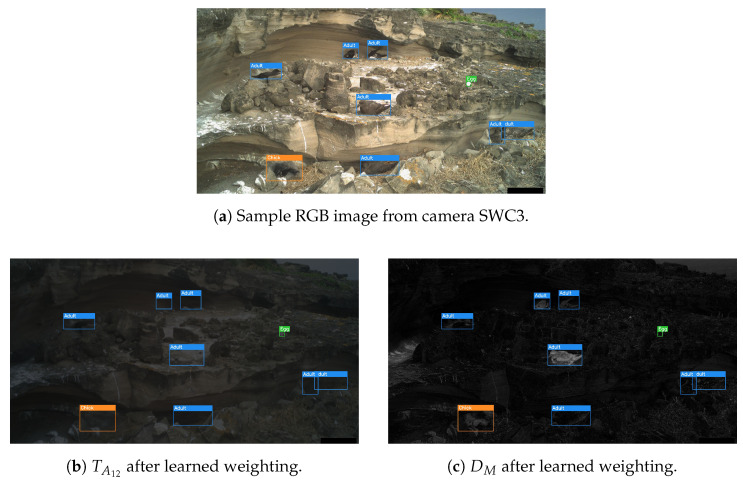
Visualisation of the $T_{A_{12}}$ (**b**) and $D_{M}$ (**c**) channels after weighting for a given image (**a**), all with ground truth annotations.

**Table 1 sensors-24-08002-t001:** Statistics on day and night image annotations and class counts for each camera.

Camera	No. Images	No. Day Images	No. Night Images	No. Adults	No. Chicks	No. Eggs
ABC1	1072	708	364	1226	535	921
ABC2	224	126	98	350	33	34
ABC3	407	188	219	694	254	34
ABC4	365	302	63	661	64	475
ABC5	149	115	34	258	85	80
SWC1	308	305	3	129	0	403
SWC2	1330	1062	268	2795	570	1198
SWC3	506	372	134	1804	389	319
SWC4	14	14	0	19	0	13
SWC5	108	59	49	162	103	0

**Table 2 sensors-24-08002-t002:** Statistics on class counts across day and night modalities for each camera.

Camera	No. Adults, Day	No. Chicks, Day	No. Eggs, Day	No. Adults, Night	No. Chicks, Night	No. Eggs, Night
ABC1	628	338	756	598	197	165
ABC2	198	11	47	152	22	9
ABC3	311	107	18	383	147	16
ABC4	461	31	463	200	33	12
ABC5	188	43	80	70	42	0
SWC1	127	0	401	2	0	2
SWC2	1807	288	1008	988	282	190
SWC3	1079	221	279	725	168	40
SWC4	19	0	13	0	0	0
SWC5	57	56	0	105	47	0

**Table 3 sensors-24-08002-t003:** Cameras selected for the training, validation, and test splits.

Set	Cameras	Images
Train	ABC1, ABC3, ABC5, SWC2, SWC4, SWC5	3080
Validation	ABC2, ABC4	589
Test	SWC1, SWC3	814

**Table 4 sensors-24-08002-t004:** Mean average precision (mAP) calculated on the validation and test sets for each method.

Method	Validation Set		Test Set	
	mAP@0.5	mAP@0.05:0.95	mAP@0.5	mAP@0.05:0.95
Baseline	0.492	0.266	0.632	0.383
$T_{A_{12}}$ + $D_{M}$ + *W*	0.543	*0.292*	**0.762**	**0.475**
$T_{A_{12}}$ + $D_{M}$ + $S_{E}$	**0.551**	**0.297**	*0.750*	*0.468*
$T_{A_{12}}$ + $D_{M}$	0.516	0.275	0.739	0.464
$T_{A_{12}}$	0.518	0.285	0.721	0.447

$T_{A_{12}}$: temporal average 12 $D_{M}$: difference mask *W*: fixed channel weighting $S_{E}$: Squeeze-and-Excitation channel weighting. Highest performance is denoted in **bold** and second-highest in *italics*.

**Table 5 sensors-24-08002-t005:** Class average precision (AP) values on the test set of our best method compared to the baseline method.

Model	AP@0.5			AP@0.05:0.95		
	Adult	Chick	Egg	Adult	Chick	Egg
Baseline	0.8	0.406	0.69	0.526	0.282	0.341
$T_{A_{12}}$ + $D_{M}$ + *W*	0.879 (+9.9%)	0.65 (+60.1%)	0.758 (+9.9%)	0.581 (+10.5%)	0.454 (+61.0%)	0.391 (+14.7%)

**Table 6 sensors-24-08002-t006:** Comparison of computational cost of the baseline and the two best methods: $T_{A_{12}}$ + $D_{M}$ + *W* and $T_{A_{12}}$ + $D_{M}$ + $S_{E}$.

Method	Training Time (ms/Batch)	Inference Time (ms/Batch)	GPU Memory (GB)
Baseline	501	197	16.5
$T_{A_{12}}$ + $D_{M}$ + *W*	610 (+21.8%)	272 (+38.1%)	17.2 (+4.24%)
$T_{A_{12}}$ + $D_{M}$ + $S_{E}$	703 (+40.3%)	293 (+48.7%)	17.4 (+5.45%)

## Data Availability

The dataset presented in this article is not readily available because it is part of an ongoing multi-partner collaborative study. Requests to access the datasets should be directed to Malcolm Nicoll (malcolm.nicoll@ioz.ac.uk).

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
