# Peer review of "Improving Object Detection for Time-Lapse Imagery Using Temporal Features in Wildlife Monitoring"

_sensors, 2024, doi:10.3390/s24248002_

Round 1
Reviewer 1 Report
Comments and Suggestions for Authors
This paper presents a novel method to enhance object detection in time-lapse imagery, specifically applied to wildlife monitoring. The authors propose leveraging temporal features from prior frames in time-lapse sequences to improve the accuracy of object detectors. By integrating two additional spatial feature channels that capture stationary and non-stationary elements of the scene, the method aims to reduce the number of false positives caused by stationary background objects. The proposed approach demonstrates a 24% improvement in mean Average Precision (mAP@0.05:0.95) over a baseline single-frame object detector. The study uses a large dataset of breeding tropical seabirds captured by camera traps, providing a compelling case for the applicability of the technique in ecological monitoring.
The paper’s application to wildlife monitoring is highly relevant, particularly in terms of automatically processing large volumes of data captured by camera traps. The study contributes to enhancing the accuracy and efficiency of monitoring animal populations, a key aspect of conservation efforts. The results show a clear improvement in the accuracy of object detection, particularly for challenging cases like detecting chicks, which are often confused with background elements. The method of temporal feature integration is shown to outperform traditional single-frame methods and even more complex models like ConvLSTM in certain aspects.
There are some comments and suggestions below:
The methods proposed (especially with ConvLSTM and weighted channel inputs) introduce additional complexity and computational cost. While the paper acknowledges this, a more detailed discussion on the scalability of these methods to larger datasets or real-time applications would be useful, particularly for conservation efforts in large-scale wildlife monitoring projects.
For the application of lightweight CNN models for object detection tasks, more relevant works should be discussed in the introduction part, such as: https://doi.org/10.1007/s00170-022-10335-8
Although the method shows significant improvements on the RI petrel dataset, it would be helpful to see a broader evaluation across different types of wildlife and other camera trap datasets to demonstrate the generalizability of the proposed techniques.
It would be valuable to compare the proposed temporal feature approach with other methods that also integrate temporal information in object detection, such as methods based on attention mechanisms or sequence-based models like transformers. This would provide a clearer picture of how the proposed approach stands against current state-of-the-art temporal models.
A more detailed analysis of the computational requirements, including memory usage, training times, and inference times, particularly when scaled to larger datasets, would provide useful insights for researchers looking to apply these methods in real-world scenarios.
Evaluating the proposed approach on other wildlife datasets would enhance its robustness and applicability. It would be beneficial to show that the improvements observed on the RI petrel dataset can be generalized to other species or environments.
Overall, this manuscript presents a significant contribution to the field of wildlife monitoring through the development of a novel method for improving object detection in time-lapse imagery. The method’s ability to reduce false positives and improve detection accuracy by leveraging temporal features is impressive. The experiments are well-designed, and the results provide a compelling case for the approach’s efficacy. However, there is room for improvement in terms of scalability, comparison with other temporal methods, and generalization to other datasets. Overall, I recommend the manuscript for acceptance with minor revisions.
Author Response
Dear Reviewer,
Thank you for your time, considerations and valuable comments. Please find our point by point responses below.
The Authors.
Comments 1: For the application of lightweight CNN models for object detection tasks, more relevant works should be discussed in the introduction part, such as: https://doi.org/10.1007/s00170-022-10335-8
Response 1: Thank you for this suggestion. In the related work section, we have focused on works where CNNs have been applied specifically for wildlife monitoring. For this reason, we do not believe this work is as relevant as the other referenced works.
Comments 2: It would be valuable to compare the proposed temporal feature approach with other methods that also integrate temporal information in object detection, such as methods based on attention mechanisms or sequence-based models like transformers. This would provide a clearer picture of how the proposed approach stands against current state-of-the-art temporal models.
Response 2: While this would provide a valuable comparison of our method to state-of-the-art temporal methods, this would be unrealistic given the time constraints of the review period (5 days).
Comments 3: A more detailed analysis of the computational requirements, including memory usage, training times, and inference times, particularly when scaled to larger datasets, would provide useful insights for researchers looking to apply these methods in real-world scenarios.
Response 3: Thank you, we agree that metrics on the computational cost are a valuable assessment of our tested methods. In response, we have added Section 5.1 titled “Comparison of Computational Cost” on page 15, line 332.
Comments 4: Evaluating the proposed approach on other wildlife datasets would enhance its robustness and applicability. It would be beneficial to show that the improvements observed on the RI petrel dataset can be generalized to other species or environments.
Response 4: We appreciate this comment, but this is also unrealistic in this timeframe in particular if we are to perform thorough model selection like we have for the RI petrel dataset.
Reviewer 2 Report
Comments and Suggestions for Authors
This paper presents a method for leveraging temporal information to improve object detection accuracy in camera trap images. The proposed method is novel, plausible, and clearly defined, and the authors provide an exceptional level of diligence in defining their splits. Overall I recommend acceptance.
The main strengths of this paper are that (1) the proposed method is presented in a level of detail that would be sufficient for replication, primarily in section 2.2, (2) the proposed method yields results that are definitively better than a reasonable baseline (YOLOv7), and (3) the level of diligence shown in carefully separating cameras into train/val/test splits exceeds that of any other paper I have seen in this space. The paper is well-written; I have no typographical suggestions. The dataset is quite small (~4.5k images annotated with boxes) and is not necessarily visually representative of other camera trap images, so it's hard to say how much the results presented here would generalize, but the claims made here are not excessively grand, and it can be left to the community to assess generalizability.
In the remainder of this review, I will raise minor criticisms that would suggest fairly significant changes to the text, but I consider them stylistic recommendations, so I am not indicating a required revision.
Firstly, I would consider dropping the entire ConvLSTM experiment. The paper already has a clear and reasonable baseline (YOLOv7); by adding the ConvLSTM experiment, the authors create an additional experimental condition that is neither clear nor particularly reasonable: it's a somewhat arbitrary approach that is not particularly expected to work well, and is not given the same level of attention to hyperparameter tuning that the other methods are, so it's not interesting or informative that it performs poorly. It also appears at a point in the paper that is a bit precarious in term of clarity, as I'll describe below, and dropping section 2.3 would help the overall flow. I am not making this a requirement for my recommendation of acceptance, and I know it's sometimes hard to drop something that you invested time in, but I encourage the authors to take an objective look at whether this is really helping to communicate the thesis of the paper. For example, although I'm not recommending this, it would be *much* more interesting to compare to alternative YOLO versions than to include the ConvLSTM experiment. The choice of YOLOv7 is arbitrary, and the choice of YOLOv7-W6 is even more arbitrary, which is all fine; you have to pick some reasonable baseline, and this paper is not the place to compare every possible YOLO architecture. But comparing to other other YOLO architectures - at the very least, other YOLOv7 architectures - would be far more informative to the reader than comparing to ConvLSTM.
Second, the paper became very hard to follow when section 2.2.3 arrived. I was able to put it all together eventually, but I'm a fairly well-informed reader, and I almost completely lost the narrative thread in section 2.2.3. The core of the issue, I think, is that the idea of generating two new input channels and modifying the YOLOv7 architecture to add those two input channels is quite intuitive, and at the end of section 2.2.2, the reader assumes this is what's going to happen. Sections 2.2.1 and 2.2.2 describe those new input channels very clearly. But then section 2.2.3 starts, and a set of weights are introduced that seem like they should just be learned as part of the modified, five-channel-input YOLOv7 architecture. The description in section 2.2.3 is complete, so after a few passes I was able to follow it, but even now I'm still not entirely sure why this isn't just learned as part of the optimization. Two small changes would help a lot here IMO: (1) at the beginning of section 2.2.3, address this issue directly, i.e. assume that the reader thinks you're going to just add two extra input channels to the YOLOv7 architecture, and explain why that's *not* what you do, i.e. why it's necessary to learn weights outside of the normal optimization process, and (2) add a *lot* more detail to the caption of Figure 2. In that caption, remind the reader what the boxes are, remind the reader what Fsq/Fex/Fscale are, etc.
Third, and less important, the colour correction steps are not clearly motivated. I can imagine why this is important, but it would help the reader if you added a couple of sentences about why this is necessary.
Fourth, I would consider moving all of the YOLOv7 hyperparameter optimization and data augmentation details to an appendix. This has very little to do with your contribution, and it interrupts the narrative with a number of equations that are more distracting than helpful. For example, listing the learning rate schedule of OneCycle is unnecessary IMO; this is a property of YOLOv7 that isn't important to this paper. This is OK in an appendix, although even there, I don't think that level of detail about the learning rate schedule is necessary, a citation would be sufficient.
Lastly, a suggestion that is outside the scope of my review and has no impact on my recommendation: the authors have a really interesting approach here, but I see no indication that the implementation has been released publicly. Historical precedent shows that the odds of anyone adopting a method in the AI for conservation/ecology space that does not start from a reference implementation provided by the authors are very close to zero percent. Consequently, the odds of anyone using this method in real ecological practice would go up by an extraordinary amount if the authors released their method as an open-source implementation. It's not even necessary IMO to release any model weights, although that wouldn't hurt, but the gap between theory and practice is *much* easier to bridge if the authors release their training and inference code, and even easier to bridge if the authors are able to release some sample data, or loosely replicate the work on data that is already public (even if the quantitative results are not the same).
Author Response
Dear Reviewer,
Thank you for your time, considerations and valuable comments. Please find our point by point responses below.
The Authors.
Comments 1: Firstly, I would consider dropping the entire ConvLSTM experiment.  The paper already has a clear and reasonable baseline (YOLOv7); by adding the ConvLSTM experiment, the authors create an additional experimental condition that is neither clear nor particularly reasonable: it's a somewhat arbitrary approach that is not particularly expected to work well, and is not given the same level of attention to hyperparameter tuning that the other methods are, so it's not interesting or informative that it performs poorly.  It also appears at a point in the paper that is a bit precarious in term of clarity, as I'll describe below, and dropping section 2.3 would help the overall flow.
Response 1: Thank you for this suggestion, we agree that ConvLSTM is perhaps not sufficiently motivated, and the lack of hyperparameter optimisation or related experiments reduces its usefulness in the context of comparison to our other methods. Therefore, we have decided to remove Section 2.3 and remove all references to ConvLSTM as suggested by the Reviewer.
Comments 2: For example, although I'm not recommending this, it would be *much* more interesting to compare to alternative YOLO versions than to include the ConvLSTM experiment.  The choice of YOLOv7 is arbitrary, and the choice of YOLOv7-W6 is even more arbitrary, which is all fine; you have to pick some reasonable baseline, and this paper is not the place to compare every possible YOLO architecture.  But comparing to other other YOLO architectures - at the very least, other YOLOv7 architectures - would be far more informative to the reader than comparing to ConvLSTM.
Response 2: We appreciate this suggestion. However, our use of ConvLSTM was as a supplementary layer to YOLOv7 to provide an additional input channel that captures temporal features that is learned (rather than replacing YOLOv7). This experiment was to act as a comparison of a deep-learned temporal feature channel against our hand-crafted temporal feature channels. Regardless, we do agree that evaluating other architectures would be useful, since it is our hypothesis that our method should work for other, related object detection architectures to YOLOv7. To prove this hypothesis rigorously, however, would be unrealistic given the timeframe of this review period (5 days).
Comments 3: Second, the paper became very hard to follow when section 2.2.3 arrived.  I was able to put it all together eventually, but I'm a fairly well-informed reader, and I almost completely lost the narrative thread in section 2.2.3. 
Sections 2.2.1 and 2.2.2 describe those new input channels very clearly.  But then section 2.2.3 starts, and a set of weights are introduced that seem like they should just be learned as part of the modified, five-channel-input YOLOv7 architecture.
Response 3: Thank you, we agree that the narrative in Section 2.2.3 does not flow well from Section 2.2.2, and the use of channel weightings has not explained motivation to justify its usage in our proposed method. Therefore, we have changed the introductory text in Section 2.2.3 (line 138) to improve the flow, and we have explained our reasoning for using these learned weightings in lines 140 to 145.
Comments 4: (2) add a *lot* more detail to the caption of Figure 2.  In that caption, remind the reader what the boxes are, remind the reader what Fsq/Fex/Fscale are, etc.
Response 4: We have added more detail to thoroughly describe Fsq, Fex and Fscale in the caption of this figure, which is now Figure 3 on page 6.
Comments 5: Third, and less important, the colour correction steps are not clearly motivated.  I can imagine why this is important, but it would help the reader if you added a couple of sentences about why this is necessary.
Response 5: We agree. We have added an explanation for the use of colour correction in Section 2.2.2 on lines 133 to 136. Additionally, we have added Figure 2 after line 136 which illustrates the effect of colour correction visually on the Temporal Average and Difference Mask.
Comments 6: Fourth, I would consider moving all of the YOLOv7 hyperparameter optimization and data augmentation details to an appendix.  This has very little to do with your contribution, and it interrupts the narrative with a number of equations that are more distracting than helpful.  For example, listing the learning rate schedule of OneCycle is unnecessary IMO; this is a property of YOLOv7 that isn't important to this paper. This is OK in an appendix, although even there, I don't think that level of detail about the learning rate schedule is necessary, a citation would be sufficient.
Response 6: Thank you for this comment, we have moved the hyperparameter optimisation (Section 4.4) to Appendix Section A.1 on page 17. We have also removed the equation for the OneCycle LR in this section and added a reference to its paper instead (reference 17).
Comments 7: Lastly, a suggestion that is outside the scope of my review and has no impact on my recommendation: the authors have a really interesting approach here, but I see no indication that the implementation has been released publicly.  Historical precedent shows that the odds of anyone adopting a method in the AI for conservation/ecology space that does not start from a reference implementation provided by the authors are very close to zero percent.  Consequently, the odds of anyone using this method in real ecological practice would go up by an extraordinary amount if the authors released their method as an open-source implementation.  It's not even necessary IMO to release any model weights, although that wouldn't hurt, but the gap between theory and practice is *much* easier to bridge if the authors release their training and inference code, and even easier to bridge if the authors are able to release some sample data, or loosely replicate the work on data that is already public (even if the quantitative results are not the same).
Response 7: We appreciate this suggestion. We have added a brief sentence in Section 2, page 3, line 84 which links to the GitHub repository. This repository is currently empty, but we will shortly provide the code and instructions for its usage (after this review period).